# Novel and Potent Acetylcholinesterase Inhibitors for the Treatment of Alzheimer’s Disease from Natural (±)-7,8-Dihydroxy-3-methyl-isochroman-4-one

**DOI:** 10.3390/molecules27103090

**Published:** 2022-05-11

**Authors:** Xinnan Li, Yilin Jia, Junda Li, Pengfei Zhang, Tiantian Li, Li Lu, Hequan Yao, Jie Liu, Zheying Zhu, Jinyi Xu

**Affiliations:** 1State Key Laboratory of Natural Medicines, Department of Medicinal Chemistry, China Pharmaceutical University, Nanjing 211198, China; xinnanli@126.com (X.L.); ammie10@163.com (Y.J.); zgykdxljd0938@163.com (J.L.); zpf834220218@163.com (P.Z.); 2Division of Molecular Therapeutics & Formulation, School of Pharmacy, The University of Nottingham, University Park Campus, Nottingham NG7 2RD, UK; paytl4@nottingham.ac.uk (T.L.); Li.Lu@nottingham.ac.uk (L.L.); Zheying.Zhu@nottingham.ac.uk (Z.Z.); 3Department of Organic Chemistry, China Pharmaceutical University, Nanjing 211198, China

**Keywords:** Alzheimer’s disease, (±)-7,8-dihydroxy-3-methyl-isochroman-4-one, acetylcholinesterase inhibitors, antioxidant activity, molecular docking

## Abstract

Alzheimer’s disease (AD) is a neurodegenerative disease that causes memory and cognitive decline as well as behavioral problems. It is a progressive and well recognized complex disease; therefore, it is very urgent to develop novel and effective anti-AD drugs. In this study, a series of novel isochroman-4-one derivatives from natural (±)-7,8-dihydroxy-3-methyl-isochroman-4-one [(±)-XJP] were designed and synthesized, and their anti-AD potential was evaluated. Among them, compound **10a** [(*Z*)-3-acetyl-1-benzyl-4-((6,7-dimethoxy-4-oxoisochroman-3-ylidene)methyl)pyridin-1-ium bromide] possessed potent anti-acetylcholinesterase (AChE) activity as well as modest antioxidant activity. Further molecular modeling and kinetic investigations revealed that compound **10a** was a dual-binding inhibitor that binds to both catalytic anionic site (CAS) and peripheral anionic site (PAS) of the enzyme AChE. In addition, compound **10a** exhibited low cytotoxicity and moderate anti-A*β* aggregation efficacy. Moreover, the in silico screening suggested that these compounds could pass across the blood–brain barrier with high penetration. These findings show that compound **10a** was a promising lead from a natural product with potent AChE inhibitory activity and deserves to be further developed for the prevention and treatment of AD.

## 1. Introduction

Alzheimer’s disease (AD) is a neurodegenerative disease and the most common type of dementia with the symptoms of progressive memory loss, cognitive impairment, changes of personality, and linguistic disorders [1,2,3]. There are currently more than 50 million AD patients worldwide, with an estimated increase to 139 million patients by 2050, leading to a significant financial burden on families and society [4,5,6,7,8,9].

The senile plaque (SP) generated by *β*-amyloid (A*β*) and neurofibrillary tangles (NFTs) made of phosphorylated tau proteins in the hippocampus are considered as the main pathogenic etiologies of AD [10]. The pathological mechanism of AD is complex, and several hypotheses have been proposed, including cholinergic injury hypothesis [11,12], *β*-amyloid protein cascade hypothesis [13], tau protein hyperphosphorylation hypothesis [14], metal ion poisoning hypothesis [15], calcium imbalance hypothesis [16], chronic inflammation hypothesis [17], and oxygen free radical damage hypothesis [18]. Among these hypotheses, the cholinergic injury is the most widely accepted and its clinical effectiveness has been approved. In the human brain, acetylcholine functions as a cholinergic neurotransmitter and plays an important role in cognition, learning, and memory. Over the past decades, many studies have found that a huge number of cholinergic neurons in the brains of AD patients are damaged, and the level of acetylcholine during nerve impulse transmission is greatly reduced, affecting signal transmission and leading to learning and cognitive failure [19,20,21,22]. Compelling evidence supports the concept that the shortage of acetylcholine is one of the main causes of AD, indicating that elevating acetylcholine level would improve AD patients [23].

AChE inhibitors, such as donepezil, rivastigmine, and galantamine, are the most commonly used anti-AD drugs clinically (Figure 1A) [1]. The AChE inhibitors work by binding the catalytic active site (CAS) at the bottom of AChE, leading to the increased level of acetylcholine (ACh). Recent research has found that the peripheral active site (PAS), which is located at the entrance of the AChE canyon active site, is linked to the neurotoxic cascade of AD caused by acetylcholine hydrolysis and AChE-induced A*β* aggregation [24]. Thus, dual interaction of binding sites both on CAS and PAS would lead to a better inhibitory activity against AChE for the treatment of AD [25,26,27]. (±)-7,8-Dihydroxy-3-methyl-isochroman-4-one [(±)-XJP] (Figure 1B) is a novel and structurally distinct natural polyphenolic molecule that our group isolated from the peel of a banana (*Musa sapientum* L.) and which displays potent antihypertensive and antioxidant activities (Figure 1B) [28,29].

The indenone fragment of donepezil occupies PAS of AChE. Due to the structural similarity between indenone and isochromanone, the indenone fragment of donepezil was cleverly designed by replacing it with the isochromanone structure of (±)-XJP. We hoped that the isochromanone structure could occupy PAS as well. The benzyl piperidine fragment of donepezil is the key pharmacophore that occupies CAS, and, combined with our previous studies [30,31,32], we retained the cyclic structure of this fragment and replaced the piperidine ring with a pyridine ring. The central region of the AChE active pocket becomes narrower and consequently restricts the passage of ring structures containing larger substituents; the bottom region of the pocket becomes relatively larger and therefore contains several lateral cavities that could be further occupied to improve the affinity of a compound on AChE (Figure 2B). In a previous work [31], we fused (±)-XJP with a donepezil analog possessing a pyridine ring to obtain a series of new compounds with no substituents on the pyridine ring, and the most potent compound had good AChE inhibitory activity with IC_50_ value of 21.1 ± 0.8 nM. On this basis, when designing new molecules, we introduced acetyl or aminocarbonyl groups that would occupy less space on the pyridine ring for the following reasons: the smaller acyl structure does not affect benzyl pyridine fragment to access CAS; and the spatial configuration of acyl group and hydrogen binding donors or acceptors that they contain has the potential to form more interactions with the active pocket. In our previous study, the structural modification of (±)-XJP provided an analog of it, named (±)-XJP-B, which displayed more potent activity than natural (±)-XJP. Therefore, by fusing the pharmacophores of (±)-XJP-B and donepezil, we designed and synthesized thirty-five new 4-isochromanone derivatives and evaluated their AChE inhibition to investigate the potential for AD treatment (Figure 2A).

## 2. Results and Discussions

### 2.1. Synthesis

The synthesis of new isochroman-4-one derivatives **10a**–**10s** and **13a**–**13p** is depicted in Figure 1. Briefly, the synthetic route is divided into two parts: the first part is synthesis of key intermediate **7**, and the second part is synthesis of the target compounds. Firstly, commercially available **1** was reduced under the action of sodium borohydride to obtain intermediate **2**, then intermediate **2** was etherified with tert-butyl bromoacetate under two-phase environment and alkaline conditions to obtain intermediate **3**. In the presence of a strong base, alcoholysis of ester and hydrolysis of ester occurred successively to produce intermediate **4**. Under the action of oxalyl chloride, **4** produced the corresponding acid chloride, which was reacted with *N*,*O*-dimethylhydroxylamine hydrochloride to yield Weinreb amide intermediate **5**. In an anhydrous, oxygen-free and low temperature environment, **5** underwent intramolecular condensation under the action of tert-butyl lithium to obtain XJP derivative **6**; intermediate **6** was condensed with bromopyridinecarboxaldehyde under high temperature conditions to obtain key intermediate **7**. The bromine substituent on the pyridine ring of intermediate **7** could be reacted with tributyl(1-ethoxyethylene) tin under the catalysis of bis-dibenzylideneacetone palladium to obtain intermediate **8,** placed in a strong acid environment, which can be converted to intermediate **9** through enols; finally, with different substituted benzyl bromide under high temperature conditions to generate target compounds **10a**–**10s**. Under the catalysis of palladium acetate and xantphos phosphorus ligand, the bromine group on intermediate **7** was replaced by 2,4,6-trichlorophenyl formate to form ester intermediate **11**. In a closed environment of high temperature and pressure, **11** led to the aminolysis reaction of ester yielding intermediate **12**. Similarly, it reacted with different substituted benzyl bromides in dried acetonitrile under reflux to yield the target compounds **13a**–**13p**. All target compounds were characterized by ^1^H NMR, ^13^C NMR, and HR-ESI-MS.

### 2.2. In Vitro Inhibition of AChE Enzyme and Structure-Activity Relationship

The inhibitory potencies of compounds **10a**–**10s** and **13a**–**13p** toward AChE were evaluated using the Ellman’s assay [23]. Donepezil was used as a positive control. The anti-AChE activities for all target compounds expressed as IC_50_ values were summarized in Table 1. The results in Table 1 indicated that most of the target compounds exhibited potent inhibitory activities against AChE. Overall, fourteen compounds had better inhibitory activities than positive control donepezil (IC_50_ = 12.06 nM). Among them, compound **10a** (IC_50_ = 1.61 nM) was the most potent compound in the first series of compounds **10a**–**10s,** showing the best inhibitory activity of all the target compounds. Compound **13b** (IC_50_ = 3.54 nM) was the best compound in the second series of compounds **13a**–**13p**. Analyzing the results of activity from the perspective of substituents on the benzyl group, it could be found that the substitution of different groups and positions on the benzene ring has a significant differentiation on the inhibitory activity. On the one hand, based on the substituted position in benzene ring, the order of inhibitory potency against AChE was: ortho > meta > para. Furthermore, for the different substituents at the same position, the activity order of the first series of compounds was H > F > Me > Cl > Br > NO_2_ > CF_3_, and the activity order of the second series was F > H > Me > Cl > Br > CF_3_; it could be seen that the more appropriate substituents were H (**10a**) or F (**13b**), and the less active substituents were CF_3_ (**10s** and **13p**) in these two series, respectively, indicating the order of activity was not much related to the electrical effect of the substituent groups and it may be also related to the steric effect of substituent groups. By analysis of the substituents on the pyridine ring, we found that the first series of compounds **10a**–**10s** performed better in activity overall. In summary, the above results suggested that the acetyl substitution on the pyridine ring (**10a**–**10s**) was better than the carbamoyl substitution (**13a**–**13p**), and the carbonyl moiety may be a key group in the series of skeleton structures. We chose the most potent compound **10a** for further investigations.

### 2.3. Kinetic Study of AChE Inhibition

In order to analyze the AChE inhibitory mechanism of these compounds, the most potent compound **10a** was selected for kinetic testing. The mechanistic inhibition study of compound **10a** on the AChE was carried out through kinetic parameters such as maximal velocity (Vmax), inhibitory concentration (Ki) and Michalis–Menten dissociation constant (Km). The reciprocal Lineweaver–Burk plot of compound **10a** on AChE demonstrated that as the concentration of inhibitor increased, both slopes and intercepts increased, and the lines generated by compound **10a** with different concentrations eventually converged in the second quadrant (Figure 3). The results in Figure 3 clearly demonstrated that compound **10a** is a mixed-type inhibitor for AChE, suggesting that compound **10a** bound both CAS and PAS of AChE. 

### 2.4. In Vitro Cytotoxicity of Compounds **10a** and **13b**

To evaluate the cytotoxicity of these new compounds, cell viability was tested on SH-SY5Y cells. The representative compounds **10a** and **13b** in these two series were selected for cytotoxicity. As shown in Figure 4, both compounds **10a** and **13b** did not significantly alter cell viability at concentrations of 10, 25 and 50 μM, indicating that compounds **10a** and **13b** had no neurotoxicity up to the concentration at 50 μM.

### 2.5. In Vitro Antioxidant Activity of Compound **10a**

DPPH assay was used to evaluate the free radical scavenging ability of the representative compound **10a**, while Trolox was used as a control [33]. As shown in Table 2, compound **10a** displayed moderate radical scavenging activity (31.89% ± 3.23% at 50 μM). The result showed that compound **10a** possessed moderate antioxidant activity at a concentration of 50 μM.

### 2.6. Inhibition of Aβ Self-Aggregation of Compound **10a**

Aggregation of *β*-amyloid peptide (A*β*) is one of the main histopathological hallmarks of AD [34]. Therefore, we selected the optimal compound **10a** using a thioflavin T assay to determine the prevention of A*β* aggregation. As shown in Table 3, compound **10a** had a moderate inhibitory effect on the A*β* aggregation, with an inhibitory rate of 56.95 ± 2.68% at a concentration of 25 μM. The result showed that compound **10a** exhibited moderate anti-A*β* aggregation efficacy.

### 2.7. Molecular Docking

In order to assess how ligand interacts with enzyme, docking simulations were performed using the Glide program of Schrodinger. The X-ray crystal structure of human AChE (PDB entry 4EY7) was downloaded from PDB. Based on the in vitro inhibitory results, compound **10a** was selected as a typical ligand for the evaluation. The docking results are shown in Figure 5. The isochromanone fragment was located at the PAS site outside pocket, and the benzyl-substituted pyridine fragment was located at the CAS site inside pocket. Some previous studies showed a proposed H-bonding of donepezil’s C=O group with human AChE; however, we did not find the proposed H-bonding of C=O group of compound **10a** with human AChE, but we found that the pharmacophore of compound **10a** based on isochromanone structure of ±(XJP-B) could occupy PAS well. Among them, the benzene ring of the isochromanone fragment forms a π-π interaction with Trp-286 residue, the pyridine ring forms a π-π interaction with Tyr-337 residue, and the benzyl ring formed a π-π interaction with Trp-86 residue (brown-dashed line); the acetyl group on the pyridine ring formed a 2.2 Å hydrogen bond with Tyr-124 residue (black-dashed line); the N atom on pyridine ring was connected to Try-337 residue and Trp-86 residue, the two aromatic rings of the radical formed three π-cations interaction (blue-dashed line). All these results indicated that compound **10a** can simultaneously bind to the PAS and CAS of AChE, which is in accordance with the results of the kinetic study.

### 2.8. Prediction of Toxicity and Drug-Likeness Properties

To evaluate the drug-likeness characteristics and predict toxicity of the target compounds, blood–brain barrier (BBB) penetration, human intestinal absorption (HIA), caco-2 cells permeability, AMES mutagenesis, and rat acute toxicity were calculated using admetSAR web-based application tool [35]. Some compounds with better activity were selected for this experiment. As shown in Table 4, according to the predicted values for BBB penetration, all compounds would be able to penetrate into the CNS. However, in the case of HIA and Caco-2 permeability, all compounds performed less well. Compound **10n** with an interposition nitro substitution on the benzyl group showed the only exception to the AMES mutagenicity, which showed potential genotoxicity (“+” sign), indicating that nitro was not a suitable substituent. Overall, the results indicated that these new compounds exhibit good drug-like properties, but there was scope for further improvement.

## 3. Experimental Section

### 3.1. Chemistry

#### 3.1.1. Materials and Methods

All commercially available chemicals and reagents were analytical grade and used without further purification only if otherwise noted. The ^1^H NMR and ^13^C NMR were recorded on Bruker-300 spectrometers (Bruker Company, Karlsruhe, Germany) using TMS as an internal standard. High-resolution mass spectrometry was accomplished on an Agilent 1100-LC-MSD-Trap/SL mass spectrometer (Agilent Technologies Inc., Santa Clara, CA, USA). Huanghai HSGF 254 silica gel plates (Yantai, China) were used for TLC. Silica gel 60 H (200–300 mesh), general chromatography was performed manufactured by Qingdao Haiyang Chemical Group Co., Ltd. (Qingdao, China).

#### 3.1.2. Synthesis of (2-Bromo-4,5-dimethoxyphenyl)methanol (**2**) 

To a solution of 2-bromo-4,5-dimethoxybenzaldehyde (**1**) (81.61 mmol, 1 eq) in 100 mL methanol, sodium borohydride (40.80 mmol, 0.5 eq) was slowly added at 0 °C and the mixture was stirred at the same temperature. After the completion of the reaction was detected by TLC, crushed ice was added into the mixture and stirred for another 1 h. After filtration, the filtrate was washed with cold water and collected as a white solid [32].

#### 3.1.3. Synthesis of Tert-butyl 2-((2-bromo-4,5-dimethoxybenzyl)oxy)acetate (**3**)

To a solution of (2-bromo-4,5-dimethoxyphenyl)methanol (**2**) (40.47 mmol, 1 eq) in 100 mL toluene, *tert*-butyl bromoacetate (48.57 mmol, 1.2 eq), fresh 40% KOH aqueous solution (100 mL) and the catalytic amount of tetrabutylammonium bromide were added successively, stirred at 50 °C for 1 h. After the completion of the reaction was detected by TLC, the mixture was extracted with ethyl acetate (3 × 200 mL). The combined organic layers were washed with brine (200 mL) and dried over anhydrous Na_2_SO_4_. The organic layers were removed in vacuo to give crude colorless oil **3** [32].

#### 3.1.4. Synthesis of 2-((2-Bromo-4,5-dimethoxybenzyl)oxy)acetic Acid (**4**)

To a solution of *tert*-butyl 2-((2-bromo-4,5-dimethoxybenzyl)oxy)acetate (**3**) in 100 mL methanol, a solution of sodium methanol (101.04 mmol, 2.5 eq) in methanol was slowly added; the mixture was stirred at room temperature for about 15 min, and an appropriate amount of H_2_O was slowly added to the mixture with vigorous stirring for 5 min. After the reaction was completed, the solvent was concentrated and the aqueous layer was extracted with ethyl acetate (3 × 200 mL). The pH of the aqueous layer was adjusted to 2 by slowly adding concentrated hydrochloric, and then the aqueous layer was extracted three times with ethyl acetate. The combined organic layers were washed with brine (100 mL) and dried over anhydrous Na_2_SO_4_. The organic layers were removed in vacuo to give crude white solid **4** [32].

#### 3.1.5. Synthesis of 2-((2-Bromo-4,5-dimethoxybenzyl)oxy)-n-methoxy-n-methylacetamide (**5**)

Intermediate2-((2-bromo-4,5-dimethoxybenzyl)oxy)acetic acid (**4**) (32.77 mmol, 1 eq) was dissolved in dry DCM (50 mL), oxalyl chloride (98.32 mmol, 3 eq)and 1–2 drops of DMF solution were slowly added and the reaction stirred 20 min at room temperature. The organic layers were removed in vacuo to give crude acyl chloride intermediate. In another flask, 50 mL of dry acetonitrile was used as solvent. *N,O*-dimethylhydroxylamine hydrochloride (36.05 mmol, 1.1 eq) and anhydrous potassium carbonate (81.93 mmol, 2.5 eq) were added sequentially and mixed well. After the reaction completed, the mixture was filtered to remove the residual insoluble matter, and the filtrate was concentrated and purified by column chromatography to give a white solid **5** [32].

#### 3.1.6. Synthesis of 6,7-Dimethoxyisochroman-4-one (**6**)

Intermediate 2-((2-bromo-4,5-dimethoxybenzyl)oxy)-*N*-methoxy-*N*-methylacetamide **5** (14.36 mmol, 1 eq) was dissolved in anhydrous THF under an N_2_ atmosphere and placed at a low temperature of −78 °C. A solution of 1.3 M *tert*-butyllithium (35.90 mmol, 2.5 eq) in *n*-pentane was added and the mixture was stirred for another 10 min. At the end of the reaction, a saturated aqueous ammonium chloride solution was added to quench the reaction and the mixture was brought to room temperature. The residue was extracted with ethyl acetate (EA) (3 × 200 mL), the organic layers were combined, washed with brine, dried over anhydrous Na_2_SO_4_, and the organic layers were concentrated and purified by column chromatography to give pale yellow solid **6**. ^1^H NMR (300 MHz, CDCl_3_) *δ* 7.50 (s, 1H), 6.64 (s, 1H), 4.85 (s, 2H), 4.33 (s, 2H), 3.95 (d, *J* = 6.7 Hz, 6H). ^13^C NMR (75 MHz, CDCl_3_) *δ* 196.7, 154.0, 149.1, 127.5, 126.3, 118.8, 109.3, 81.1, 72.7, 56.2, 56.1. MS (ESI) *m/z*: 209.18 [M + H]^+^.

#### 3.1.7. Synthesis of (Z)-3-((3-Bromopyridin-4-yl)methylene)-6,7-dimethoxyisochroman-4-one (**7**)

To a solution of Intermediate **6** (4.80 mmol, 1 eq) in dry toluene, 3-bromopyridine-4-aldehyde (7.20 mmol, 1.5 eq) and *p*-toluenesulfonic acid monohydrate (7.20 mmol, 1.5 eq) were added sequentially and the mixture stirred at reflux for 4 h. At the end of the reaction, the reaction system was cooled to room temperature, and the pH was adjusted to 8 by adding saturated sodium bicarbonate. The mixture was extracted with dichloromethane (3 × 200 mL). The organic layer was washed with brine and dried over anhydrous Na_2_SO_4_, and purified by column chromatography to give yellow solid **7**. ^1^H NMR (300 MHz, CDCl_3_) δ 8.76 (s, 1H), 8.50 (d, J = 5.1 Hz, 1H), 8.06 (d, J = 5.0 Hz, 1H), 7.58 (s, 1H), 7.25 (s, 1H), 6.66 (s, 1H), 5.29 (s, 2H), 3.99 (d, J = 4.8 Hz, 6H). ^13^C NMR (75 MHz, CDCl_3_) δ 178.5, 157.5, 156.0, 149.6, 149.6, 148.2, 147.1, 132.5, 125.7, 118.6, 118.6, 118.4, 106.2, 102.6, 70.5, 56.1, 56.0. MS (ESI) m/z:376.41 [M + H]^+^.

#### 3.1.8. Synthesis of (Z)-3-((3-(1-Ethoxyvinyl)pyridine-4-yl)methylene)-6,7-dimethoxyisochroman-4-one (**8**)

To a solution of intermediate 7 (3.46 mmol, 1 eq) in dry toluene at nitrogen atmosphere, *p*-(diaziridine acetonide)palladium (0.05 mmol, 0.04 eq), tributyl(1-ethoxyethylene)tin (1.37 mmol, 1.2 eq) and triphenylphosphine (0.09 mmol, 0.08 eq) were added sequentially and stirred at reflux for 10 h, then brought to room temperature. After filtration, the filtrate was concentrated and purified by column chromatography to give yellow solid, which was further recrystallized by dichloromethane/petroleum ether to give intermediate **8**.

#### 3.1.9. Synthesis of (Z)-3-((3-Acetylpyridin-4-yl)methylene)-6,7-dimethoxyisochroman-4-one (**9**)

Intermediate **8** dissolved in 6 N aqueous HCl and stirred for 1 h at 50 °C. After the reaction completed, the hydrochloric acid was neutralized by adding saturated sodium bicarbonate solution and the organic layer was extracted with dichloromethane. The organic layer was washed with saturated salt water, dried over anhydrous sodium sulphate, concentrated and recrystallized using a mixture of dichloromethane/petroleum to give yellow solid **9**.

#### 3.1.10. Synthesis of 2,4,6-Trichlorophenyl 4-((5,6-dimethoxy-1-oxo-2,3-dihydro-1H-inden-2-yl)methyl)nicotinate (**11**)

To a solution of intermediate **7** (3.46 mmol, 1 eq) in dry toluene at nitrogen atmosphere, xantphos (0.34 mmol, 0.1 eq), palladium acetate (0.17 mmol, 0.05 eq) and triethylamine (6.91 mmol, 2 eq) were added sequentially and stirred at 100 °C for 5 min, then brought to room temperature. 2,4,6-Trichlorophenyl ester (5.18 mmol, 1.5 eq) was added to the mixture. After filtration, the filtrate was concentrated and purified by column chromatography to give a yellow solid, which was further recrystallized by dichloromethane/petroleum to give intermediate **11**.

#### 3.1.11. Synthesis of (Z)-4-((6,7-Dimethoxy-4-oxoisochroman-3-ylidene)methyl)nicotinamide (**12**)

Intermediate **11** was added into 0.4 M ammonia (17.3 mmol, 5 eq) in dioxane solution in a pressure resistant sealing tube. The mixture was stirred at 80 °C for 17 h. The solvent was removed in vacuo, and was further recrystallized by ethyl acetate to give intermediate **12**.

#### 3.1.12. Synthesis of Target Compounds **10a**–**10s**

To a solution of intermediate **9** in dry acetonitrile, benzyl bromide with different substituents (0.71 mmol, 3 eq) was added and stirred for 1–3 h at 85 °C. The solvent was removed under reduced pressure and recrystallized by ethyl acetate to give **10a**–**10s**.

(Z)-3-Acetyl-1-benzyl-4-((6,7-dimethoxy-4-oxoisochroman-3-ylidene)methyl)pyridin-1-ium bromide (**10a**)

Yellow solid, 90.5% yield. ^1^H NMR (300 MHz, DMSO-*d*_6_) *δ* 9.75 (s, 1H), 9.10 (d, *J* = 6.4 Hz, 1H), 8.69 (d, *J* = 6.8 Hz, 1H), 7.60 (s, 2H), 7.51–7.36 (m, 4H), 7.31 (s, 1H), 7.12 (s, 1H), 5.89 (s, 2H), 5.50 (s, 2H), 3.88 (d, *J* = 14.1 Hz, 6H), 2.76 (s, 3H). ^13^C NMR (75 MHz, DMSO-*d*_6_) *δ* 197.8, 176.9, 156.9, 155.5, 150.9, 149.7, 148.9, 145.3, 144.7, 134.9, 129.7, 129.4, 128.1, 127.1, 125.8, 125.2, 121.2, 116.4, 107.6, 105.0, 103.5, 67.6, 62.9, 56.8, 56.2, 30.7. HR-MS (ESI) *m/z*: calcd for C_26_H_24_NO_5_ [M]^+^ 430.1649, found 430.1649.

(Z)-3-Acetyl-4-((6,7-dimethoxy-4-oxoisochroman-3-ylidene)methyl)-1-(2-fluorobenzyl)pyridin-1-ium Bromide (**10b**)

Yellow solid, 85.3% yield. ^1^H NMR (300 MHz, DMSO-*d*_6_) *δ* 9.62 (s, 1H), 9.04 (d, *J* = 6.8 Hz, 1H), 8.77 (d, *J* = 6.7 Hz, 1H), 7.65 (dd, *J* = 13.7, 6.2 Hz, 1H), 7.61–7.53 (m, 1H), 7.47 (s, 1H), 7.41 (d, *J* = 10.0 Hz, 1H), 7.36 (d, *J* = 4.2 Hz, 2H), 7.17 (s, 1H), 6.00 (s, 2H), 5.56 (s, 2H), 3.95 (s, 3H), 3.90 (s, 3H), 2.77 (s, 3H). ^13^C NMR (75 MHz, DMSO-*d*_6_) *δ* 197.7, 176.8, 162.2, 159.7, 156.6, 155.5, 149.7, 149.2, 145.9, 145.5, 135.3, 135.1, 132.5, 131.8, 128.0, 125.7, 121.8, 121.2, 116.4, 107.6, 103.3, 67.6, 57.6, 56.8, 56.2, 30.6. HR-MS (ESI) *m/z*: calcd for C_27_H_26_FNO_6_ [M + CH_3_OH]^+^ 448.1817, found 480.1823.

(Z)-3-Acetyl-4-((6,7-dimethoxy-4-oxoisochroman-3-ylidene)methyl)-1-(3-fluorobenzyl)pyridin-1-ium Bromide (**10c**)

Yellow solid, 86.4% yield. ^1^H NMR (300 MHz, DMSO-*d*_6_) *δ* 9.69 (s, 1H), 9.11 (d, *J* = 6.8 Hz, 1H), 8.73 (d, *J* = 6.7 Hz, 1H), 7.60–7.51 (m, 2H), 7.50–7.43 (m, 2H), 7.37–7.27 (m, 2H), 7.15 (s, 1H), 5.91 (s, 2H), 5.53 (s, 2H), 3.93 (s, 3H), 3.88 (s, 3H), 2.77 (s, 3H). ^13^C NMR (75 MHz, DMSO-*d*_6_) *δ* 197.8, 176.8, 164.3, 161.0, 156.5, 155.5, 149.7, 149.1, 146.0, 145.3, 137.2, 135.1, 131.7, 128.1, 125.6, 121.2, 116.6, 116.3, 107.9, 107.6, 103.4, 67.6, 62.1, 56.8, 56.2, 30.6. HR-MS (ESI) *m/z*: calcd for C_27_H_26_FNO_6_ [M + CH_3_OH]^+^ 480.1817, found 480.1819.

(Z)-3-Acetyl-4-((6,7-dimethoxy-4-oxoisochroman-3-ylidene)methyl)-1-(4-fluorobenzyl)pyridin-1-ium Bromide (**10d**)

Yellow solid, 88.3% yield. ^1^H NMR (300 MHz, DMSO-*d*_6_) *δ* 9.70 (s, 1H), 9.11 (d, *J* = 6.7 Hz, 1H), 8.72 (d, *J* = 6.7 Hz, 1H), 7.75 (dd, *J* = 8.2, 5.7 Hz, 2H), 7.45 (s, 1H), 7.35 (dd, *J* = 10.7, 6.8 Hz, 3H), 7.16 (s, 1H), 5.90 (s, 2H), 5.53 (s, 2H), 3.94 (s, 3H), 3.89 (s, 3H), 2.78 (s, 3H). ^13^C NMR (75 MHz, DMSO-*d*_6_) *δ* 197.8, 176.8, 164.7, 161.4, 156.4, 155.5, 149.7, 148.9, 145.7, 145.1, 135.2, 135.1, 132.1, 131.0, 128.1, 121.2, 116.7, 116.4, 107.9, 107.5, 103.4, 67.5, 62.0, 56.8, 56.2, 30.6. HR-MS (ESI) *m/z*: calcd for C_27_H_26_FNO_6_ [M + CH_3_OH]^+^ 480.1817, found 480.1822.

(Z)-3-Acetyl-1-(2-chlorobenzyl)-4-((6,7-dimethoxy-4-oxoisochroman-3-ylidene)methyl)pyridin-1-ium Bromide (**10e**)

Brown solid, 87.9% yield. ^1^H NMR (300 MHz, DMSO-*d*_6_) *δ* 9.60 (s, 1H), 8.97 (d, *J* = 6.8 Hz, 1H), 8.77 (d, *J* = 6.8 Hz, 1H), 7.66 (d, *J* = 7.8 Hz, 1H), 7.58–7.52 (m, 1H), 7.49 (d, *J* = 6.4 Hz, 2H), 7.46 (s, 1H), 7.38 (s, 1H), 7.16 (s, 1H), 6.02 (s, 2H), 5.55 (s, 2H), 3.94 (s, 3H), 3.89 (s, 3H), 2.75 (s, 3H). ^13^C NMR (75 MHz, DMSO-*d*_6_) *δ* 197.8, 176.8, 156.8, 155.5, 149.7, 149.3, 146.3, 145.6, 135.1, 133.4, 132.1, 131.7, 131.4, 130.5, 128.6, 127.9, 121.2, 112.9, 107.9, 107.6, 103.2, 67.6, 60.9, 56.8, 56.2, 30.6. HR-MS (ESI) *m/z*: calcd for C_26_H_23_ClNO_5_ [M]^+^ 464.1259, found 464.1254.

(Z)-3-Acetyl-1-(3-chlorobenzyl)-4-((6,7-dimethoxy-4-oxoisochroman-3-ylidene)methyl)pyridin-1-ium Bromide (**10f**)

Yellow solid, 90.0% yield. ^1^H NMR (300 MHz, DMSO-*d*_6_) *δ* 9.70 (s, 1H), 9.11 (d, *J* = 6.8 Hz, 1H), 8.73 (d, *J* = 6.7 Hz, 1H), 7.80 (s, 1H), 7.61 (s, 1H), 7.55 (s, 2H), 7.45 (s, 1H), 7.35 (s, 1H), 7.16 (s, 1H), 5.90 (s, 2H), 5.54 (s, 2H), 3.93 (s, 3H), 3.89 (s, 3H), 2.78 (s, 3H). ^13^C NMR (75 MHz, DMSO-*d*_6_) *δ* 197.8, 176.8, 156.5, 155.5, 149.7, 149.1, 146.0, 145.2, 136.9, 135.2, 135.1, 134.1, 131.5, 129.8, 129.4, 128.2, 128.1, 121.2, 107.9, 107.5, 103.4, 67.5, 62.0, 56.8, 56.2, 30.7. HR-MS (ESI) *m/z*: calcd for C_26_H_23_ClNO_5_ [M]^+^ 464.1259, found 464.1257.

(Z)-3-Acetyl-1-(4-chlorobenzyl)-4-((6,7-dimethoxy-4-oxoisochroman-3-ylidene)methyl)pyridin-1-ium Bromide (**10g**)

Brown solid, 90.3% yield. ^1^H NMR (300 MHz, DMSO-*d*_6_) *δ* 9.67 (s, 1H), 9.07 (d, *J* = 6.6 Hz, 1H), 8.70 (d, *J* = 6.8 Hz, 1H), 7.66 (d, *J* = 8.5 Hz, 2H), 7.55 (d, *J* = 8.5 Hz, 2H), 7.42 (s, 1H), 7.32 (s, 1H), 7.13 (s, 1H), 5.88 (s, 2H), 5.51 (s, 2H), 3.91 (s, 3H), 3.86 (s, 3H), 2.75 (s, 3H). ^13^C NMR (75 MHz, DMSO-*d*_6_) *δ* 197.8, 176.8, 156.4, 155.5, 149.7, 149.0, 145.9, 145.2, 135.2, 135.1, 134.6, 133.6, 131.5, 131.5, 129.6, 129.6, 128.1, 121.2, 107.9, 107.6, 103.4, 67.6, 62.0, 56.8, 56.2, 30.7. HR-MS (ESI) *m/z*: calcd for C_26_H_23_ClNO_5_ [M]^+^ 464.1259, found 464.1247.

(Z)-3-Acetyl-1-(2-bromobenzyl)-4-((6,7-dimethoxy-4-oxoisochroman-3-ylidene)methyl)pyridin-1-ium Bromide (**10h**)

Brown solid, 88.3% yield. ^1^H NMR (300 MHz, DMSO-*d*_6_) *δ* 9.59 (s, 1H), 8.93 (d, *J* = 7.0 Hz, 1H), 8.76 (d, *J* = 6.8 Hz, 1H), 7.80 (d, *J* = 7.9 Hz, 1H), 7.54–7.47 (m, 1H), 7.47–7.41 (m, 2H), 7.39–7.31 (m, 2H), 7.14 (s, 1H), 5.96 (s, 2H), 5.54 (s, 2H), 3.92 (s, 3H), 3.87 (s, 3H), 2.73 (s, 3H). ^13^C NMR (75 MHz, DMSO-*d*_6_) *δ* 197.7, 176.8, 156.8, 156.1, 155.5, 154.9, 153.4, 149.7, 149.4, 135.1, 135.0, 133.8, 131.3, 130.1, 129.1, 127.9, 123.6, 121.2, 107.9, 107.6, 103.2, 68.5, 63.1, 56.8, 56.2, 30.6. HR-MS (ESI) *m/z*: calcd for C_26_H_23_BrNO_5_ [M]^+^ 508.0754, found 508.0761.

(Z)-3-Acetyl-1-(3-bromobenzyl)-4-((6,7-dimethoxy-4-oxoisochroman-3-ylidene)methyl)pyridin-1-ium Bromide (**10i**)

Yellow solid, 80.5% yield. ^1^H NMR (300 MHz, DMSO-*d*_6_) *δ* 9.65 (s, 1H), 9.08 (d, *J* = 6.8 Hz, 1H), 8.71 (d, *J* = 6.8 Hz, 1H), 7.90 (s, 1H), 7.69–7.59 (m, 2H), 7.48–7.40 (m, 2H), 7.33 (s, 1H), 7.13 (s, 1H), 5.86 (s, 2H), 5.51 (s, 2H), 3.91 (s, 3H), 3.86 (s, 3H), 2.75 (s, 3H). ^13^C NMR (75 MHz, DMSO-*d*_6_) *δ* 197.8, 176.8, 156.5, 155.5, 149.7, 149.1, 146.0, 145.2, 137.1, 135.2, 135.1, 132.7, 132.3, 131.8, 128.6, 128.1, 122.7, 121.2, 107.9, 107.6, 103.4, 67.6, 62.0, 56.8, 56.2, 30.7. HR-MS (ESI) *m/z*: calcd for C_27_H_26_BrNO_6_ [M + CH_3_OH]^+^ 540.1016, found 540.1018.

(Z)-3-Acetyl-1-(4-bromobenzyl)-4-((6,7-dimethoxy-4-oxoisochroman-3-ylidene)methyl)pyridin-1-ium Bromide (**10j**)

Brown solid, 88.7% yield. ^1^H NMR (300 MHz, DMSO-*d*_6_) *δ* 9.65 (s, 1H), 9.08 (d, *J* = 6.5 Hz, 1H), 8.72 (d, *J* = 6.5 Hz, 1H), 7.72 (d, *J* = 7.3 Hz, 2H), 7.60 (d, *J* = 7.8 Hz, 2H), 7.46 (s, 1H), 7.35 (s, 1H), 7.15 (s, 1H), 5.87 (s, 2H), 5.53 (s, 2H), 3.94 (s, 3H), 3.89 (s, 4H), 2.77 (s, 3H). ^13^C NMR (75 MHz, DMSO-*d*_6_) *δ* 197.8, 176.8, 156.4, 155.5, 149.7, 149.0, 145.9, 145.2, 135.2, 135.1, 134.0, 132.6, 132.6, 131.7, 131.7, 128.1, 123.3, 121.2, 107.9, 107.6, 103.4, 67.5, 62.1, 56.8, 56.2, 30.6. HR-MS (ESI) *m/z*: calcd for C_26_H_23_BrNO_5_ [M]^+^ 508.0754, found 508.0767.

(Z)-3-Acetyl-4-((6,7-dimethoxy-4-oxoisochroman-3-ylidene)methyl)-1-(2-methylbenzyl)pyridin-1-ium Bromide (**10k**)

Brown solid, 84.9% yield. ^1^H NMR (300 MHz, DMSO-*d*_6_) *δ* 9.59 (s, 1H), 8.88 (d, *J* = 6.8 Hz, 1H), 8.76 (d, *J* = 6.7 Hz, 1H), 7.46 (s, 1H), 7.39 (d, *J* = 6.5 Hz, 3H), 7.32 (t, *J* = 6.9 Hz, 1H), 7.24–7.14 (m, 2H), 5.94 (s, 2H), 5.54 (s, 2H), 3.94 (s, 3H), 3.89 (s, 3H), 2.76 (s, 3H), 2.39 (s, 3H). ^13^C NMR (75 MHz, DMSO-*d*_6_) *δ* 197.8, 176.8, 156.5, 155.5, 149.7, 149.0, 146.1, 145.3, 137.3, 135.1, 135.1, 132.8, 131.3, 129.7, 129.3, 128.0, 127.1, 121.2, 107.8, 107.6, 103.3, 67.6, 61.2, 56.8, 56.2, 30.7, 19.4. HR-MS (ESI) *m/z*: calcd for C_28_H_29_NO_6_ [M + CH_3_OH]^+^ 476.2068, found 476.2076.

(Z)-3-Acetyl-4-((6,7-dimethoxy-4-oxoisochroman-3-ylidene)methyl)-1-(3-methylbenzyl)pyridin-1-ium Bromide (**10l**)

Yellow solid, 85.4% yield. ^1^H NMR (300 MHz, DMSO-*d*_6_) *δ* 9.69 (s, 1H), 9.09 (d, *J* = 6.8 Hz, 1H), 8.73 (d, *J* = 6.7 Hz, 1H), 7.48–7.38 (m, 4H), 7.35 (d, *J* = 4.0 Hz, 1H), 7.29 (d, *J* = 6.8 Hz, 1H), 7.16 (s, 1H), 5.83 (d, *J* = 17.7 Hz, 2H), 5.53 (s, 2H), 3.94 (s, 3H), 3.89 (s, 3H), 2.79 (s, 3H), 2.36 (s, 3H). ^13^C NMR (75 MHz, DMSO-*d*_6_) *δ* 197.8, 176.8, 156.4, 155.5, 149.6, 148.9, 145.7, 145.2, 139.0, 135.2, 135.1, 134.6, 130.4, 129.9, 129.6, 128.0, 126.4, 121.2, 107.9, 107.6, 103.4, 67.5, 62.9, 56.8, 56.2, 30.7, 21.4. HR-MS (ESI) *m/z*: calcd for C_28_H_29_NO_6_ [M + CH_3_OH]^+^ 476.2068, found 476.2076.

(Z)-3-Acetyl-4-((6,7-dimethoxy-4-oxoisochroman-3-ylidene)methyl)-1-(4-methylbenzyl)pyridin-1-ium Bromide (**10m**)

Yellow solid, 89.9% yield. ^1^H NMR (300 MHz, DMSO-*d*_6_) *δ* 9.67 (s, 1H), 9.08 (d, *J* = 6.9 Hz, 1H), 8.71 (d, *J* = 6.7 Hz, 1H), 7.54 (d, *J* = 7.9 Hz, 2H), 7.45 (s, 1H), 7.38–7.22 (m, 3H), 7.16 (s, 1H), 5.84 (s, 2H), 5.53 (s, 2H), 3.94 (s, 3H), 3.89 (s, 3H), 2.78 (s, 3H), 2.35 (s, 3H). ^13^C NMR (75 MHz, DMSO-*d*_6_) *δ* 197.8, 176.8, 156.3, 155.5, 149.6, 148.9, 145.6, 145.1, 139.4, 135.2, 135.1, 131.8, 130.2, 130.2, 129.4, 129.4, 128.1, 121.2, 107.9, 107.5, 103.5, 67.5, 62.7, 56.8, 56.2, 30.7, 21.2. HR-MS (ESI) *m/z*: calcd for C_28_H_29_NO_6_ [M + CH_3_OH]^+^ 476.2068, found 476.2075.

(Z)-3-Acetyl-4-((6,7-dimethoxy-4-oxoisochroman-3-ylidene)methyl)-1-(2-nitrobenzyl)pyridin-1-ium Bromide (**10n**)

Yellow solid, 89.2% yield. ^1^H NMR (300 MHz, DMSO-*d*_6_) *δ* 9.59 (s, 1H), 9.01 (d, *J* = 7.0 Hz, 1H), 8.82 (d, *J* = 6.7 Hz, 1H), 8.32 (d, *J* = 7.2 Hz, 1H), 7.85 (q, *J* = 6.7 Hz, 1H), 7.82–7.72 (m, 1H), 7.45 (d, *J* = 12.7 Hz, 2H), 7.27 (d, *J* = 7.3 Hz, 1H), 7.18 (s, 1H), 6.27 (s, 2H), 5.58 (s, 2H), 3.95 (s, 3H), 3.90 (s, 3H), 2.74 (s, 3H). ^13^C NMR (75 MHz, DMSO-*d*_6_) *δ* 197.7, 176.8, 156.8, 155.5, 149.7, 149.5, 147.9, 146.8, 145.9, 135.4, 135.1, 135.0, 130.8, 130.5, 129.9, 128.0, 126.0, 121.2, 107.9, 107.6, 103.3, 67.6, 60.5, 56.8, 56.2, 30.4. HR-MS (ESI) *m/z*: calcd for C_27_H_26_N_2_O_8_ [M + CH_3_OH]^+^ 507.1762, found 507.1769.

(Z)-3-Acetyl-4-((6,7-dimethoxy-4-oxoisochroman-3-ylidene)methyl)-1-(3-nitrobenzyl)pyridin-1-ium Bromide (**10o**)

Yellow solid, 87.3% yield. ^1^H NMR (300 MHz, DMSO-*d*_6_) *δ* 9.73 (s, 1H), 9.17 (d, *J* = 6.8 Hz, 1H), 8.74 (d, *J* = 6.7 Hz, 1H), 8.63 (s, 1H), 8.34 (d, *J* = 8.2 Hz, 1H), 8.13 (d, *J* = 7.7 Hz, 1H), 7.81 (t, *J* = 8.0 Hz, 1H), 7.45 (s, 1H), 7.35 (s, 1H), 7.16 (s, 1H), 6.05 (s, 2H), 5.54 (s, 2H), 3.94 (s, 3H), 3.89 (s, 3H), 2.78 (s, 3H). ^13^C NMR (75 MHz, DMSO-*d*_6_) *δ* 197.8, 176.8, 156.5, 155.5, 149.7, 149.1, 148.5, 146.1, 145.3, 136.4, 136.4, 135.2, 135.1, 131.2, 128.1, 124.9, 124.7, 121.2, 107.8, 107.6, 103.4, 67.6, 61.7, 56.8, 56.2, 30.7. HR-MS (ESI) *m/z*: calcd for C_27_H_26_N_2_O_8_ [M + CH_3_OH]^+^ 507.1762, found 507.1768.

(Z)-3-Acetyl-4-((6,7-dimethoxy-4-oxoisochroman-3-ylidene)methyl)-1-(4-nitrobenzyl)pyridin-1-ium Bromide (**10p**)

Brown solid, 91.0% yield. ^1^H NMR (300 MHz, DMSO-*d*_6_) *δ* 9.69 (s, 1H), 9.12 (d, *J* = 6.9 Hz, 1H), 8.76 (d, *J* = 6.7 Hz, 1H), 8.35 (d, *J* = 8.6 Hz, 2H), 7.86 (d, *J* = 8.6 Hz, 2H), 7.46 (s, 1H), 7.37 (s, 1H), 7.16 (s, 1H), 6.06 (s, 2H), 5.55 (s, 2H), 3.94 (s, 3H), 3.89 (s, 3H), 2.77 (s, 3H). ^13^C NMR (75 MHz, DMSO-*d*_6_) *δ* 197.7, 176.8, 156.6, 155.5, 149.7, 149.3, 148.3, 146.22, 145.5, 141.7, 135.2, 135.1, 130.6, 130.6, 128.1, 124.6, 124.6, 121.2, 107.9, 107.5, 103.3, 67.6, 61.8, 56.8, 56.2, 30.6. HR-MS (ESI) *m/z*: calcd for C_27_H_26_N_2_O_8_ [M + CH_3_OH]^+^ 507.1762, found 507.1771.

(Z)-3-Acetyl-4-((6,7-dimethoxy-4-oxoisochroman-3-ylidene)methyl)-1-(2-(trifluoromethyl)benzyl)pyridin-1-ium Bromide (**10q**)

Yellow solid, 90.9% yield. ^1^H NMR (300 MHz, DMSO-*d*_6_) *δ* 9.57 (s, 1H), 8.92 (d, *J* = 7.0 Hz, 1H), 8.77 (d, *J* = 6.8 Hz, 1H), 7.94 (d, *J* = 7.0 Hz, 1H), 7.81–7.73 (m, 1H), 7.70 (t, *J* = 7.5 Hz, 1H), 7.44 (s, 1H), 7.38 (s, 1H), 7.26 (d, *J* = 7.3 Hz, 1H), 7.14 (s, 1H), 6.12 (s, 2H), 5.55 (s, 2H), 3.92 (s, 3H), 3.87 (s, 3H), 2.73 (s, 3H). ^13^C NMR (75 MHz, DMSO-*d*_6_) *δ* 197.8, 176.8, 156.9, 155.5, 149.7, 149.5, 146.7, 145.9, 135.1, 135.1, 134.1, 132.2, 130.4, 130.2, 128.0, 127.3, 126.9, 126.3, 121.2, 107.9, 107.6, 103.1, 67.6, 59.8, 56.8, 56.2, 30.6. HR-MS (ESI) *m/z*: calcd for C_28_H_26_F_3_NO_6_ [M + CH_3_OH]^+^ 530.1785, found 530.1784.

(Z)-3-Acetyl-4-((6,7-dimethoxy-4-oxoisochroman-3-ylidene)methyl)-1-(3-(trifluoromethyl)benzyl)pyridin-1-ium Bromide (**10r**)

Yellow solid, 87.9% yield. ^1^H NMR (300 MHz, DMSO-*d*_6_) *δ* 9.73 (s, 1H), 9.14 (d, *J* = 6.5 Hz, 1H), 8.75 (d, *J* = 5.1 Hz, 1H), 8.13 (s, 1H), 7.95 (d, *J* = 7.4 Hz, 1H), 7.87 (d, *J* = 7.5 Hz, 1H), 7.75 (t, *J* = 7.5 Hz, 1H), 7.46 (s, 1H), 7.37 (s, 1H), 7.17 (s, 1H), 6.00 (s, 2H), 5.55 (s, 2H), 3.94 (s, 3H), 3.90 (s, 3H), 2.79 (s, 3H). ^13^C NMR (75 MHz, DMSO-*d*_6_) *δ* 197.8, 176.8, 156.5, 155.5, 149.7, 149.1, 146.1, 145.3, 135.9, 135.2, 135.1, 133.7, 130.8, 130.3, 129.9, 128.1, 126.5, 126.2, 121.2, 107.9, 107.6, 103.4, 67.6, 62.0, 56.8, 56.2, 30.7. HR-MS (ESI) *m/z*: calcd for C_28_H_26_F_3_NO_6_ [M + CH_3_OH]^+^ 530.1785, found 530.1794.

(Z)-3-Acetyl-4-((6,7-dimethoxy-4-oxoisochroman-3-ylidene)methyl)-1-(4-(trifluoromethyl)benzyl)pyridin-1-ium Bromide (**10s**)

Brown solid, 90.9% yield. ^1^H NMR (300 MHz, DMSO-*d*_6_) *δ* 9.67 (s, 1H), 9.09 (d, *J* = 7.0 Hz, 1H), 8.72 (d, *J* = 6.7 Hz, 1H), 7.86 (d, *J* = 8.3 Hz, 2H), 7.80 (d, *J* = 8.3 Hz, 2H), 7.43 (s, 1H), 7.34 (s, 1H), 7.13 (s, 1H), 5.98 (s, 2H), 5.52 (s, 2H), 3.91 (s, 3H), 3.86 (s, 3H), 2.75 (s, 3H). ^13^C NMR (75 MHz, DMSO-*d*_6_) *δ* 197.7, 176.8, 156.5, 155.5, 149.7, 149.2, 146.1, 145.4, 139.2, 135.2, 135.1, 130.2, 130.2, 129.8, 128.1, 126.5, 126.5, 121.2, 118.8, 107.9, 107.6, 103.4, 67.6, 62.1, 56.8, 56.2, 30.6. HR-MS (ESI) *m/z*: calcd for C_28_H_26_F_3_NO_6_ [M + CH_3_OH]^+^ 530.1785, found 530.1794.

#### 3.1.13. Synthesis of Target Compounds **13a**–**13p**

To a solution of intermediate **12** in dry acetonitrile, benzyl bromide with different substituents (0.71 mmol, 3 eq) was added and stirred for 1–3 h at 85 °C. The solvent was removed under reduced pressure and recrystallized by ethyl acetate to give **13a**–**13p**.

(Z)-1-Benzyl-3-carbamoyl-4-((6,7-dimethoxy-4-oxoisochroman-3-ylidene)methyl)pyridin-1-ium Bromide (**13a**)

Red solid, 93.1% yield. ^1^H NMR (300 MHz, DMSO-*d*_6_) *δ* 9.37 (s, 1H), 9.09 (d, *J* = 6.1 Hz, 1H), 8.77 (d, *J* = 6.5 Hz, 1H), 8.50 (s, 1H), 8.32 (s, 1H), 7.62 (s, 2H), 7.49 (s, 3H), 7.44 (s, 1H), 7.18 (s, 1H), 7.10 (s, 1H), 5.84 (s, 2H), 5.58 (s, 2H), 3.94 (s, 3H), 3.89 (s, 3H). ^13^C NMR (75 MHz, DMSO-*d*_6_) *δ* 176.7, 165.3, 157.1, 155.5, 149.7, 147.8, 144.4, 143.2, 135.4, 135.1, 134.8, 129.8, 129.7, 129.7, 129.4, 129.4, 126.8, 121.1, 107.8, 107.6, 102.0, 67.7, 62.8, 56.8, 56.2. HR-MS (ESI) *m/z*: calcd for C_25_H_23_N_2_O_5_ [M]^+^ 431.1601, found 431.1606.

(Z)-3-Carbamoyl-4-((6,7-dimethoxy-4-oxoisochroman-3-ylidene)methyl)-1-(2-fluorobenzyl)pyridin-1-ium Bromide (**13b**)

Brown solid, 86.5% yield. ^1^H NMR (300 MHz, DMSO-*d*_6_) *δ* 9.19 (s, 1H), 8.96 (d, *J* = 7.1 Hz, 1H), 8.76 (d, *J* = 6.7 Hz, 1H), 8.44 (s, 1H), 8.26 (s, 1H), 7.62 (t, *J* = 7.4 Hz, 1H), 7.56–7.49 (m, 1H), 7.41 (s, 1H), 7.36 (d, *J* = 7.3 Hz, 1H), 7.32 (d, *J* = 4.8 Hz, 1H), 7.14 (s, 1H), 7.06 (s, 1H), 5.89 (s, 2H), 5.56 (s, 2H), 3.91 (s, 3H), 3.85 (s, 3H). ^13^C NMR (75 MHz, DMSO-*d*_6_) *δ* 176.7, 165.3, 163.8, 162.6, 159.3, 157.3, 155.5, 149.7, 148.0, 144.7, 143.3, 135.4, 132.5, 131.9, 126.7, 125.8, 121.7, 121.1, 116.3, 107.6, 101.9, 67.7, 63.7, 56.8, 56.2. HR-MS (ESI) *m/z*: calcd for C_25_H_22_FN_2_O_5_ [M]^+^ 449.1507, found 449.1511.

(Z)-3-Carbamoyl-4-((6,7-dimethoxy-4-oxoisochroman-3-ylidene)methyl)-1-(3-fluorobenzyl)pyridin-1-ium Bromide (**13c**)

Red solid, 90.2% yield. ^1^H NMR (300 MHz, DMSO-*d*_6_) *δ* 9.33 (s, 1H), 9.07 (d, *J* = 6.9 Hz, 1H), 8.74 (d, *J* = 6.7 Hz, 1H), 8.45 (s, 1H), 8.27 (s, 1H), 7.52 (dd, *J* = 7.8, 5.8 Hz, 2H), 7.45 (d, *J* = 8.0 Hz, 1H), 7.42 (s, 1H), 7.34–7.26 (m, 1H), 7.15 (s, 1H), 7.08 (s, 1H), 5.82 (s, 2H), 5.56 (s, 2H), 3.91 (s, 3H), 3.86 (s, 3H). ^13^C NMR (75 MHz, DMSO-*d*_6_) *δ* 176.7, 165.3, 164.3, 157.2, 155.5, 149.7, 148.0, 144.5, 143.3, 137.2, 135.3, 135.1, 131.7, 126.8, 125.6, 121.1, 116.6, 116.3, 107.8, 101.9, 92.5, 67.7, 62.0, 56.8, 56.2. HR-MS (ESI) *m/z*: calcd for C_25_H_22_FN_2_O_5_ [M]^+^ 449.1507, found 449.1510.

(Z)-3-Carbamoyl-4-((6,7-dimethoxy-4-oxoisochroman-3-ylidene)methyl)-1-(4-fluorobenzyl)pyridin-1-ium Bromide (**13d**)

Red solid, 88.3% yield. ^1^H NMR (300 MHz, DMSO-*d*_6_) *δ* 9.27 (s, 1H), 9.02 (d, *J* = 6.5 Hz, 1H), 8.72 (d, *J* = 6.8 Hz, 1H), 8.45 (s, 1H), 8.27 (s, 1H), 7.68 (dd, *J* = 8.6, 5.4 Hz, 2H), 7.41 (s, 1H), 7.31 (t, *J* = 8.8 Hz, 2H), 7.12 (s, 1H), 7.06 (s, 1H), 5.77 (s, 2H), 5.54 (s, 2H), 3.91 (s, 3H), 3.85 (s, 3H). ^13^C NMR (75 MHz, DMSO-*d*_6_) *δ* 176.7, 165.3, 161.4, 157.1, 155.5, 149.7, 147.8, 144.3, 143.2, 135.3, 135.1, 132.1, 132.0, 131.0, 126.8, 121.1, 116.7, 116.4, 107.9, 107.6, 102.0, 67.7, 62.0, 56.8, 56.2. HR-MS (ESI) *m/z*: calcd for C_25_H_22_FN_2_O_5_ [M]^+^ 449.1507, found 449.1510.

(Z)-3-Carbamoyl-1-(2-chlorobenzyl)-4-((6,7-dimethoxy-4-oxoisochroman-3-ylidene)methyl)pyridin-1-ium Bromide (**13e**)

Red solid, 89.3% yield. ^1^H NMR (300 MHz, DMSO-*d*_6_) *δ* 9.20 (s, 1H), 8.94 (d, *J* = 6.6 Hz, 1H), 8.78 (d, *J* = 6.8 Hz, 1H), 8.45 (s, 1H), 8.27 (s, 1H), 7.62 (d, *J* = 7.3 Hz, 1H), 7.55–7.49 (m, 2H), 7.49–7.46 (m, 1H), 7.43 (s, 1H), 7.16 (s, 1H), 7.10 (s, 1H), 5.95 (s, 2H), 5.57 (s, 2H), 3.92 (s, 3H), 3.86 (s, 3H). ^13^C NMR (75 MHz, DMSO-*d*_6_) *δ* 176.7, 165.3, 157.3, 155.5, 149.7, 148.2, 144.8, 143.5, 140.5, 135.2, 135.1, 133.6, 132.0, 131.8, 130.5, 128.6, 126.7, 121.1, 107.9, 107.6, 101.9, 67.7, 60.8, 56.8, 56.2. HR-MS (ESI) *m/z*: calcd for C_25_H_22_ClN_2_O_5_ [M]^+^ 465.1212, found 465.1214.

(Z)-3-Carbamoyl-1-(3-chlorobenzyl)-4-((6,7-dimethoxy-4-oxoisochroman-3-ylidene)methyl)pyridin-1-ium Bromide (**13f**)

Red solid, 90.8% yield. ^1^H NMR (300 MHz, DMSO-*d*_6_) *δ* 9.33 (s, 1H), 9.08 (d, *J* = 7.0 Hz, 1H), 8.74 (d, *J* = 6.7 Hz, 1H), 8.45 (s, 1H), 8.28 (s, 1H), 7.76 (s, 1H), 7.57 (s, 1H), 7.51 (d, *J* = 5.7 Hz, 2H), 7.41 (d, *J* = 2.4 Hz, 1H), 7.15 (s, 1H), 7.08 (s, 1H), 5.81 (s, 2H), 5.56 (s, 2H), 3.91 (s, 3H), 3.86 (s, 3H). ^13^C NMR (75 MHz, DMSO-*d*_6_) *δ* 176.7, 165.3, 157.2, 155.5, 149.7, 148.0, 144.5, 143.3, 136.9, 135.3, 135.1, 134.1, 131.5, 129.8, 129.4, 128.2, 126.9, 121.1, 107.8, 107.6, 101.9, 67.7, 61.9, 56.8, 56.2. HR-MS (ESI) *m/z*: calcd for C_25_H_22_ClN_2_O_5_ [M]^+^ 465.1212, found 465.1217.

(Z)-3-Carbamoyl-1-(4-chlorobenzyl)-4-((6,7-dimethoxy-4-oxoisochroman-3-ylidene)methyl)pyridin-1-ium Bromide (**13g**)

Red solid, 90.5% yield. ^1^H NMR (300 MHz, DMSO-*d*_6_) *δ* 9.26 (s, 1H), 9.02 (d, *J* = 7.0 Hz, 1H), 8.73 (d, *J* = 6.7 Hz, 1H), 8.44 (s, 1H), 8.27 (s, 1H), 7.62 (d, *J* = 8.5 Hz, 2H), 7.54 (d, *J* = 8.5 Hz, 2H), 7.41 (s, 1H), 7.12 (s, 1H), 7.06 (s, 1H), 5.78 (s, 2H), 5.55 (s, 2H), 3.90 (s, 3H), 3.85 (s, 3H). ^13^C NMR (75 MHz, DMSO-*d*_6_) *δ* 176.7, 165.3, 157.1, 155.5, 149.7, 147.9, 144.4, 143.2, 135.3, 135.1, 134.6, 133.6, 131.5, 131.5, 129.6, 129.6, 126.8, 121.1, 107.8, 107.5, 101.9, 67.7, 61.9, 56.8, 56.2. HR-MS (ESI) *m/z*: calcd for C_25_H_22_ClN_2_O_5_ [M]^+^ 465.1212, found 465.1215.

(Z)-1-(2-Bromobenzyl)-3-carbamoyl-4-((6,7-dimethoxy-4-oxoisochroman-3-ylidene)methyl)pyridin-1-ium Bromide (**13h**)

Brown solid, 91.2% yield. ^1^H NMR (300 MHz, DMSO-*d*_6_) *δ* 9.23 (s, 1H), 8.94 (d, *J* = 7.1 Hz, 1H), 8.81 (d, *J* = 6.8 Hz, 1H), 8.49 (s, 1H), 8.31 (s, 1H), 7.82 (d, *J* = 7.8 Hz, 1H), 7.54 (t, *J* = 7.0 Hz, 1H), 7.49–7.43 (m, 2H), 7.40 (d, *J* = 7.4 Hz, 1H), 7.19 (s, 1H), 7.13 (s, 1H), 5.94 (s, 2H), 5.60 (s, 2H), 3.94 (s, 3H), 3.89 (s, 3H). ^13^C NMR (75 MHz, DMSO-*d*_6_) *δ* 176.7, 165.3, 157.4, 155.5, 149.7, 148.2, 144.9, 143.6, 135.2, 135.1, 133.8, 133.6, 1318, 131.6, 129.1, 126.7, 123.8, 121.1, 107.8, 107.6, 101.9, 67.7, 62.8, 56.8, 56.2. HR-MS (ESI) *m/z*: calcd for C_25_H_22_BrN_2_O_5_ [M]^+^ 509.0707, found 509.0705.

(Z)-1-(3-Bromobenzyl)-3-carbamoyl-4-((6,7-dimethoxy-4-oxoisochroman-3-ylidene)methyl)pyridin-1-ium Bromide (**13i**)

Yellow solid, 89.6% yield. ^1^H NMR (300 MHz, DMSO-*d*_6_) *δ* 9.35 (s, 1H), 9.10 (d, *J* = 6.8 Hz, 1H), 8.77 (d, *J* = 6.8 Hz, 1H), 8.48 (s, 1H), 8.32 (s, 1H), 7.92 (s, 1H), 7.72–7.61 (m, 2H), 7.52–7.42 (m, 2H), 7.18 (s, 1H), 7.10 (s, 1H), 5.82 (s, 2H), 5.59 (s, 2H), 3.94 (s, 3H), 3.88 (s, 3H). ^13^C NMR (75 MHz, DMSO-*d*_6_) *δ* 176.7, 165.3, 157.2, 155.5, 149.7, 148.0, 144.5, 143.3, 137.1, 135.3, 135.1, 132.7, 132.3, 131.8, 128.6, 126.9, 122.7, 121.1, 107.9, 107.6, 101.9, 67.7, 61.9, 56.8, 56.2. HR-MS (ESI) *m/z*: calcd for C_25_H_22_BrN_2_O_5_ [M]^+^ 509.0707, found 509.0705.

(Z)-1-(4-Bromobenzyl)-3-carbamoyl-4-((6,7-dimethoxy-4-oxoisochroman-3-ylidene)methyl)pyridin-1-ium Bromide (**13j**)

Yellow solid, 88.2% yield. ^1^H NMR (300 MHz, DMSO-*d*_6_) *δ* 9.33 (s, 1H), 9.07 (d, *J* = 6.9 Hz, 1H), 8.75 (d, *J* = 6.8 Hz, 1H), 8.47 (s, 1H), 8.30 (s, 1H), 7.71 (d, *J* = 8.4 Hz, 2H), 7.59 (d, *J* = 8.4 Hz, 2H), 7.43 (s, 1H), 7.17 (s, 1H), 7.09 (s, 1H), 5.81 (s, 2H), 5.58 (s, 2H), 3.93 (s, 3H), 3.88 (s, 3H). ^13^C NMR (75 MHz, DMSO-*d*_6_) *δ* 176.7, 165.3, 157.1, 155.5, 149.7, 147.9, 144.5, 143.3, 135.3, 135.1, 134.0, 132.5, 132.5, 131.7, 131.7, 126.8, 123.3, 121.1, 107.8, 107.6, 101.9, 67.7, 62.0, 56.8, 56.2. HR-MS (ESI) *m/z*: calcd for C_25_H_22_BrN_2_O_5_ [M]^+^ 509.0707, found 509.0705.

(Z)-3-Carbamoyl-4-((6,7-dimethoxy-4-oxoisochroman-3-ylidene)methyl)-1-(2-methylbenzyl)pyridin-1-ium Bromide (**13k**)

Yellow solid, 89.8% yield. ^1^H NMR (300 MHz, DMSO-*d*_6_) *δ* 9.20 (s, 1H), 8.87 (d, *J* = 7.1 Hz, 1H), 8.79 (d, *J* = 6.8 Hz, 1H), 8.50 (s, 1H), 8.31 (s, 1H), 7.45 (s, 1H), 7.37 (t, *J* = 7.1 Hz, 2H), 7.31 (d, *J* = 6.5 Hz, 1H), 7.24–7.16 (m, 2H), 7.12 (s, 1H), 5.88 (s, 2H), 5.59 (s, 2H), 3.94 (s, 3H), 3.89 (s, 3H), 2.35 (s, 3H). ^13^C NMR (75 MHz, DMSO-*d*_6_) *δ* 176.7, 165.3, 157.2, 155.5, 149.7, 147.9, 144.6, 143.3, 137.4, 135.2, 135.1, 132.7, 131.3, 129.8, 129.5, 127.1, 126.8, 121.1, 107.8, 107.6, 101.9, 67.7, 61.1, 56.8, 56.2, 19.4. HR-MS (ESI) *m/z*: calcd for C_26_H_25_N_2_O_5_ [M]^+^ 445.1758, found 445.1765.

(Z)-3-Carbamoyl-4-((6,7-dimethoxy-4-oxoisochroman-3-ylidene)methyl)-1-(3-methylbenzyl)pyridin-1-ium Bromide (**13l**)

Yellow solid, 87.6% yield. ^1^H NMR (300 MHz, DMSO-*d*_6_) *δ* 9.34 (s, 1H), 9.08 (d, *J* = 6.8 Hz, 1H), 8.76 (d, *J* = 6.7 Hz, 1H), 8.50 (s, 1H), 8.32 (s, 1H), 7.44 (s, 2H), 7.42–7.35 (m, 2H), 7.28 (d, *J* = 6.6 Hz, 1H), 7.18 (s, 1H), 7.10 (s, 1H), 5.78 (s, 2H), 5.58 (s, 2H), 3.94 (s, 3H), 3.88 (s, 3H), 2.36 (s, 3H). ^13^C NMR (75 MHz, DMSO-*d*_6_) *δ* 176.8, 165.4, 157.2, 155.6, 149.8, 147.9, 144.5, 143.2, 139.1, 135.4, 135.2, 134.7, 130.5, 130.0, 129.7, 126.9, 126.6, 121.2, 107.9, 107.7, 102.1, 67.8, 62.9, 56.9, 56.3, 21.5. HR-MS (ESI) *m/z*: calcd for C_26_H_25_N_2_O_5_ [M]^+^ 445.1758, found 445.1758.

(Z)-3-Carbamoyl-4-((6,7-dimethoxy-4-oxoisochroman-3-ylidene)methyl)-1-(4-methylbenzyl)pyridin-1-ium Bromide (**13m**)

Yellow solid, 88.1% yield. ^1^H NMR (300 MHz, DMSO-*d*_6_) *δ* 9.32 (s, 1H), 9.06 (d, *J* = 7.1 Hz, 1H), 8.75 (d, *J* = 6.8 Hz, 1H), 8.48 (s, 1H), 8.31 (s, 1H), 7.52 (d, *J* = 8.0 Hz, 2H), 7.44 (s, 1H), 7.30 (d, *J* = 7.9 Hz, 2H), 7.17 (s, 1H), 7.09 (s, 1H), 5.77 (s, 2H), 5.55 (d, *J* = 14.5 Hz, 2H), 3.94 (s, 3H), 3.88 (s, 3H), 2.34 (s, 3H). ^13^C NMR (75 MHz, DMSO-*d*_6_) *δ* 176.7, 165.3, 157.0, 155.5, 149.7, 147.7, 144.3, 143.0, 139.4, 135.4, 135.1, 131.8, 130.2, 130.2, 129.5, 129.5, 126.8, 121.1, 107.8, 107.6, 102.0, 67.7, 62.7, 56.8, 56.2, 21.2. HR-MS (ESI) *m/z*: calcd for C_26_H_25_N_2_O_5_ [M]^+^ 445.1758, found 445.1762.

(Z)-3-Carbamoyl-4-((6,7-dimethoxy-4-oxoisochroman-3-ylidene)methyl)-1-(2-(trifluoromethyl)benzyl)pyridin-1-ium Bromide (**13n**)

Yellow solid, 89.5% yield. ^1^H NMR (300 MHz, DMSO-*d*_6_) *δ* 9.24 (s, 1H), 8.91 (d, *J* = 6.8 Hz, 1H), 8.82 (d, *J* = 6.7 Hz, 1H), 8.50 (s, 1H), 8.30 (s, 1H), 7.96 (d, *J* = 7.6 Hz, 1H), 7.81 (t, *J* = 7.4 Hz, 1H), 7.73 (t, *J* = 7.3 Hz, 1H), 7.46 (s, 1H), 7.32 (d, *J* = 7.5 Hz, 1H), 7.18 (d, *J* = 7.8 Hz, 2H), 6.09 (s, 2H), 5.61 (s, 2H), 3.95 (s, 3H), 3.89 (s, 3H). ^13^C NMR (75 MHz, DMSO-*d*_6_) *δ* 176.6, 165.2, 157.4, 155.5, 149.7, 148.4, 145.1, 143.9, 135.2, 135.1, 134.1, 132.0, 130.7, 130.2, 127.3, 126.7, 126.3, 122.6, 121.1, 107.8, 107.6, 101.8, 67.7, 59.5, 56.8, 56.2. HR-MS (ESI) *m/z*: calcd for C_26_H_22_F_3_N_2_O_5_ [M]^+^ 499.1475, found 499.1479.

(Z)-3-Carbamoyl-4-((6,7-dimethoxy-4-oxoisochroman-3-ylidene)methyl)-1-(3-(trifluoromethyl)benzyl)pyridin-1-ium Bromide (**13o**)

Yellow solid, 87.4% yield. ^1^H NMR (300 MHz, DMSO-*d*_6_) *δ* 9.38 (s, 1H), 9.13 (d, *J* = 6.8 Hz, 1H), 8.78 (d, *J* = 6.7 Hz, 1H), 8.46 (s, 1H), 8.30 (s, 1H), 8.12 (s, 1H), 7.94 (d, *J* = 7.7 Hz, 1H), 7.85 (d, *J* = 7.8 Hz, 1H), 7.74 (t, *J* = 7.8 Hz, 1H), 7.44 (s, 1H), 7.17 (s, 1H), 7.11 (s, 1H), 5.92 (s, 2H), 5.58 (s, 2H), 3.94 (s, 3H), 3.88 (s, 3H). ^13^C NMR (75 MHz, DMSO-*d*_6_) *δ* 176.7, 165.3, 157.2, 155.5, 149.7, 148.0, 144.5, 143.4, 135.9, 135.3, 135.1, 133.8, 130.8, 130.3, 129.9, 126.9, 126.6, 122.6, 121.1, 107.8, 107.6, 101.9, 67.7, 62.0, 56.8, 56.2. HR-MS (ESI) *m/z*: calcd for C_26_H_22_F_3_N_2_O_5_ [M]^+^ 499.1475, found 499.1479.

(Z)-3-Carbamoyl-4-((6,7-dimethoxy-4-oxoisochroman-3-ylidene)methyl)-1-(4-(trifluoromethyl)benzyl)pyridin-1-ium Bromide (**13p**)

Brown solid, 87.9% yield. ^1^H NMR (300 MHz, DMSO-*d*_6_) *δ* 9.38 (s, 1H), 9.12 (d, *J* = 6.8 Hz, 1H), 8.79 (d, *J* = 6.7 Hz, 1H), 8.49 (s, 1H), 8.31 (s, 1H), 7.89 (d, *J* = 8.3 Hz, 2H), 7.84 (d, *J* = 8.3 Hz, 2H), 7.45 (s, 1H), 7.18 (s, 1H), 7.13 (s, 1H), 5.97 (s, 2H), 5.60 (s, 2H), 3.95 (s, 3H), 3.89 (s, 3H). ^13^C NMR (75 MHz, DMSO-*d*_6_) *δ* 176.7, 165.3, 157.2, 155.5, 149.7, 148.1, 144.7, 143.5, 139.2, 135.3, 135.1, 130.2, 130.2, 129.8, 126.9, 126.5, 126.4, 126.2, 121.1, 107.8, 107.6, 101.9, 67.7, 62.0, 56.8, 56.2. HR-MS (ESI) *m/z*: calcd for C_26_H_22_F_3_N_2_O_5_ [M]^+^ 499.1475, found 499.1484.

### 3.2. Biological Evaluation

#### 3.2.1. Inhibition of eeAChE

AChE activity was measured by the ELISA method using AChE enzyme from electrophorus electricus (eeAChE) (C3389, Merck, Kenilworth, NJ, USA) [36]. AChE solution (2 U/mL) was prepared with 0.1 M phosphate buffer (pH 8.0) firstly. Varied concentrations of test compounds (20 μL), 1 mM 5′-dithiobis (2-nitrobenzoic acid) (DTNB) solution (100 μL), and AChE solution (40 μL) were added in order to each well of a 96-well plate. After incubation for 15 min at 37 °C, the reaction was initiated by adding 20 μL of acetylthiocholine iodide (1 mM) to the mixed solution. The absorbance of each plate was instantly measured at 405 nm using a microplate reader. The inhibition of each compound was estimated using the formula (1−Ai/Ac) × 100, where Ai and Ac represent the absorbance of AChE in the presence and absence of inhibitors, respectively. Each experiment was replicated three times. Plotting curves of log concentration via percentage of inhibition, the IC_50_ values of all tested compounds were graphically analyzed (Graph Pad Prism 8.0).

#### 3.2.2. Kinetic Study of AChE Inhibition

Kinetic study of AChE inhibition was carried out using Ellman’s method [30], which is the same as the AChE assay, to determine the kinetic mechanism of chemical inhibition. The kinetic inhibition of compound **10a** was analyzed at concentrations of 0.8 and 1.6nM. Lineweaver–Burk reciprocal plots were created by graphing 1/velocity vs. 1/[substrate] at various concentrations of substrate acetylthiocholine. The software GraphPad Prism 8.0 was used to analyze the data.

#### 3.2.3. In Vitro Antioxidant Activity Assay

Compound **10a** was tested for free radical-scavenging activity using 1,1-diphenyl-2-picryl-hydrazyl (DPPH) (D9132, Merck, Kenilworth, NJ, USA) [37]. To make a 0.1 mM concentration of DPPH stock solution, DPPH was dissolved in methanol. In a 96-well plate, 10 μL of the test sample at 100 μM and 90 μL of DPPH solution were added into each well. Trolox was used as a reference. The absorbance was measured at 545 nm after 30 min of reaction at 25 °C. 

#### 3.2.4. Self-Induced A*β* Aggregation Inhibition by Thioflavin T Assay

A total of 1mg A*β* (P9001, Beyotime, Shanghai, China) was dissolved in 1 mL of hexafluoroisopropanol and aliquoted into 50 μL per tube. Then, 20 μL of DMSO was added into one tube of A*β* solution. The test compound was prepared as a 10 mM of stock solution with DMSO and diluted to 50 μM with PBS (including the control). Then, 20 μL of the test compound and 20 μL of the A*β* solution were added to a 96-well black plate, which was gently tapped to mix the solution, covered tightly and sealed with black tape to prevent evaporation of the solvent, and left in the dark at room temperature for 24 h. Then. 160 μL of THT solution at a concentration of 5 μM was added and the fluorescence absorption was measured at an excitation wavelength of 445 nm and an emission wavelength of 490 nm. The software GraphPad Prism 8.0 was used to analyze the data.

#### 3.2.5. Cytotoxicity Assay on SH-SY5Y Cells

SH-SY5Y cells were grown in a 25 cm^2^ culture flask in fresh MEM/F12 medium, supplemented with 10% fetal bovine serum, 100 U/mL penicillin, and 100 g/mL streptomycin, and incubated at 37 °C with 5% CO_2_. Cells were seeded at a density of 1 × 10^4^ cells per well in a 96-well plate. After incubation for 24 h, medium was removed. Compounds **10a** or **13b** (10, 20, 50 μM) were prepared with serum-free media and added into each well of the plate for 24 h. After 24 h treatment, 10 μL of 3-(4,5-dimethyl-2-thiazolyl)-2,5-diphenyl-2-H-tetrazolium bromide was added into each well. After 4 h of incubation at 37 °C, 100 μL of DMSO was added to dissolve the formazan crystals, and the mixture’s absorbance was measured at 490 nm under a microplate reader (FC/K3, Thermo, Waltham, MA, USA). The survival rate was estimated using the formula Ae/Ab × 100%, where Ae and Ab represent the absorbance of SH-SY5Y cells in the presence and absence of the tested drugs, respectively. Each experiment was replicated three times.

#### 3.2.6. Molecular Docking Study

The protein structure of AChE (PDB ID: 4EY7) was downloaded from Protein Data Bank (https://www.rcsb.org) (accessed on 15 January 2022). AChE was refined and an active pocket for AChE was defined using the protein preparation and receptor grid generating components of Schrodinger software. To improve tiny molecule structures, the ligand preparation module was employed. Finally, AChE was docked using the ligand docking module.

## 4. Conclusions

In conclusion, by fusing the pharmacophores of (±)-XJP-B and donepezil, a series of novel *isochroman-4-one* derivatives were designed, synthesized and evaluated as AChE inhibitors. The AChE inhibitory activity of these new compounds was overall strong, with IC_50_ values in the nanomolar range, and representative compound **10a** [(*Z*)-3-acetyl-1-benzyl-4-((6,7-dimethoxy-4-oxoisochroman-3-ylidene)methyl)pyridin-1-ium bromide] was the most potent molecule (IC_50_ = 1.61 nM). Diverse substituents, including electron-donating groups and electron-drawing groups, were introduced into different positions on the benzene ring of the target compounds to investigate the substituent effects (the electronic effect, steric effect and position of substituted groups) and systematically analyze the structure activity relationships. Further molecular docking studies revealed that compound **10a** occupies the active pocket of AChE well, forming secondary bonds with the pocket’s amino acid residues. With the results of low cytotoxicity, anti-oxidation and inhibition of self-induced A*β* aggregation, it is strongly suggested that compound **10a** is a promising lead compound to be further studied for discovery and development of novel and effective anti-AD drugs.

## Data Availability

The data presented in this study are available within the article and Appendix A.

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
