# Peer review of "Novel and Potent Acetylcholinesterase Inhibitors for the Treatment of Alzheimer’s Disease from Natural (±)-7,8-Dihydroxy-3-methyl-isochroman-4-one"

_molecules, 2022, doi:10.3390/molecules27103090_

Round 1
Reviewer 1 Report
Review of the article "Novel and Potent Acetylcholinesterase Inhibitors for the Treatment of Alzheimer's Disease from Natural (±) -7,8-Dihydroxy-3-methyl-isochroman-4-one" by Li et al.
The authors of the article took up the problem of searching for active compounds against Alzheimer's disease. The problem is extremely important, because every year there are more and more patients suffering from this ailment. In the introduction, the authors reliably present the problem of the disease, its origins, effects and drugs available on the market. In addition, the authors smoothly justify the idea of ​​research involving the search for active analogs (±)-XJP being a hybrid (±)-XJP-B and donepezil. The entire introduction is supported by a thorough 32 literature positions.
In the "Results and synthesis" section, the authors present the synthesis of 35 new derivatives of isochroman-4-one in a simple and clear manner. The reader will be pleased to sympathize with a very fluent description of a multi-stage synthesis. In the following section, the authors conclude that 16 of the synthesized derivatives have better inhibitory activity against the AChE enzyme than the reference donepezil. The authors also perform a structure-activity relationship analysis, with the result that the substituents H and F are the key to achieving high activity. The overall results are summarized in Table 1, which facilitates the comparison of the IC50 values ​​of individual compounds. The analysis of the kinetics of AChE inhibition by compound 10a (the most active of the synthesized compounds) is presented correctly. In the following article, the authors carry out in vitro cytotoxicity tests for two compounds (10a and 13b) and tests of antioxidant activity of 10a using the DPPH method. Sections 2.7 and 2.8 also undoubtedly prove the validity of the research leading to active compounds, in particular compound 10a, which deserves further research. The whole part of the discussion ends with a brief summary resulting from the discussion in the previous sections. The experimental section was written in an exemplary manner, allowing the research to be repeated by a person familiar with the art of chemical syntheses and biological research.
The authors' great care for the way the results are presented deserves special praise.
Minor remarks:
1) Please add the producers, cities and countries of all equipment.
2) Were compounds 2-5 used for further syntheses without purification, hence there was no NMR and MS analysis?
3) Why was the DPPH method chosen to test antioxidant activity? What about the other methods?
Reviewer 2 Report
Article entitled ‘Novel and Potent Acetylcholinesterase Inhibitors for the Treatment of Alzheimer's Disease from Natural (±)-7,8-Dihydroxy-3-methyl-isochroman-4-one’ submitted by Xinnan Li et al. provides information about new potential therapeutic agents useful in the therapy of Alzheimer's disease, a serious neurodegenerative disorder that affects more and more people. It is a complex global problem with still practically no effective drugs, despite recent progress in better understanding this dysfunction and testing new medications. Authors suggest that new synthesized isochroman-4-one derivatives from natural (±)-7,8-dihydroxy-3-methyl-isochroman-4-one possess anti-acetylcholinesterase and
antioxidant activity. They performed molecular modeling and kinetic investigation and the in silico screening, which led to the conclusion that a compound called 10a (……) could be a promising lead to further investigations on the development of drugs against Alzheimer's disease.
The manuscript is well prepared and written. I have no significant comments. In my opinion, the paper would be interesting for readers of Molecules, providing new knowledge to stimulate further scientific discussion and further advanced studies on the discovery of effective and safe therapies for Alzheimer's disease.
Nevertheless, minor corrections should be introduced. In particular:
-the chemical name of the promising compound 10a is needed
- redaction of Table 1
-English could be corrected
According to Guides for Authors, affiliation, text, and references should be corrected.
Reviewer 3 Report
The authors have investigated and analyzed the novel and potent acetylcholinesterase inhibitors for the treatment of Alzheimer's Disease from a natural (±)-7,8-Dihydroxy-3-methyl-isochroman-4-one. Although the authors have demonstrated a novelty, and current study have potential for publication but needs some major modifications to further make it interesting for readership of the journal. To this end, I would like to underline couple of comments related to this manuscript just in case if the authors consider my suggestions interesting for further inclusion/revision. Overall, current work has a quality and may be considered for publication to benefit the Molecules’ community.
- The authors have assumed the introduction of acetyl or aminocarbonyl groups due to the claimed reasons given in the manuscript. The reviewer would like to know this could be more justify able if an analog would have been designed and tested without acetyl or aminocarbonyl substituents. On contrary the authors did not performed aforesaid proposed designing and theoretically presented the reasons which demonstrates a lack of scientific gap or a logical point in current designing.
- Section 2.7: ±(XJP-B) was fused to donepezil. In this perspective, during the docking studies, the authors did not highlight the role of C=O group coming from ±(XJP-B) in 10a. Although some studies have shown the role of CO and propose H-bonding of donepezil’s CO group with human AChE. What was the reason to exclude CO in this investigation during docking studies? Reviewer would like to suggest revisiting this issue before submission a revised version.
- Current version of manuscript doesn’t has docking studies of analog 13b however, the authors have claimed that molecular docking studies revealed that compounds 10a and 13b were well to occupy the active pocket of AChE, forming secondary bonds with the pocket's amino acid residues. Reviewer feels that based on this statement docking studies of 13b should be included.
- Were the observed and calculated HR-MS (ESI) values of 10a same?
- Section 2.3 is not well described to understand the Lineweaver-Burk reciprocal diagram. More detail version of kinetic study of AChE inhibition of 10a should be included.
- Table 1: In vitro AChE inhibitory activity of 10b and 13b shows that both are equipotent analogs with different functionalities. A rational for such trend should be provided with special reference to SAR.
- Several linguistic and grammatical mistakes have been detected throughout the manuscript. The reviewer is left with the feeling that manuscript needs language improvement.
Round 2
Reviewer 3 Report
Response to Reviewer 3 Comments
The authors have investigated and analyzed the novel and potent acetylcholinesterase inhibitors for the treatment of Alzheimer's Disease from a natural (±)-7,8-Dihydroxy-3-methyl-isochroman-4-one. Although the authors have demonstrated a novelty, and current study have potential for publication but needs some major modifications to further make it interesting for readership of the journal. To this end, I would like to underline couple of comments related to this manuscript just in case if the authors consider my suggestions interesting for further inclusion/revision. Overall, current work has a quality and may be considered for publication to benefit the Molecules’ community.
Point 1: The authors have assumed the introduction of acetyl or aminocarbonyl groups due to the claimed reasons given in the manuscript. The reviewer would like to know this could be more justify able if an analog would have been designed and tested without acetyl or aminocarbonyl substituents. On contrary the authors did not perform aforesaid proposed designing and theoretically presented the reasons which demonstrates a lack of scientific gap or a logical point in current designing.
Response 1: Thanks for the reviewer’s indications. Although the molecule in the manuscript has some flaws, it exhibits a strong inhibition of AChE [10a (IC50 = 1.61 nM)] and a proven safety profile, demonstrating that our compound is of great interest for future anti-AD drug development, and we will incorporate the design suggestions made by the reviewers in our subsequent work.
CURRENT COMMENT:
No doubt the authors have made great continuous efforts in designing a molecule and proposed 10 a finally. However, frankly speaking such point (round1, point1) should be considered before designing scheme. Community needs proof rather than theoretical stance.
Point 2: Section 2.7: ±(XJP-B) was fused to donepezil. In this perspective, during the docking studies, the authors did not highlight the role of C=O group coming from ±(XJP-B) in 10a. Although some studies have shown the role of CO and propose H-bonding of donepezil’s CO group with human AChE. What was the reason to exclude CO in this investigation during docking studies? Reviewer would like to suggest revisiting this issue before submission a revised version.
Response 2: That's a good point! In our attempt to discover new skeleton of Acetylcholinesterase (AChE) inhibitors, we introduced ±(XJP-B) into donepezil as new AChE inhibitors mainly to obtain the good medicinal properties and reducing lateral of the skeleton. We hypothesized that, when interacting with AChE, ±(XJP-B) would occupy PAS. In molecular docking studies, it was proved that the ±(XJP-B) part of compound 10a could occupy PAS well. However, we didn't find the propose H-bonding of compound 10a’s CO group with human AChE. So, we excluded CO in this investigation during docking studies.
CURRENT COMMENT: Thank you for authors ‘reply. No such rational or hypothesis is given in manuscript on Page 2 and even in docking discussion as well. If author found as mentioned in reply (above point 2), then such rational must be added in manuscript. This will help the rest of the researchers around the globe.
Point 3: Current version of manuscript doesn’t has docking studies of analog 13b however, the authors have claimed that molecular docking studies revealed that compounds 10a and 13b were well to occupy the active pocket of AChE, forming secondary bonds with the pocket's amino acid residues. Reviewer feels that based on this statement docking studies of 13b should be included.
Response 3: Thanks for the reviewer’s indications. Because compound 10a is the most potent compound, we chose 10a as the object of molecular docking study, and we mistakenly included 13b in the conclusion section. Now we have rewritten the conclusion.
CURRENT COMMENT: Ok, thank you.
Point 4: Were the observed and calculated HR-MS (ESI) values of 10a same?
Response 4: Thank you for highlighting this detail. The observed and calculated HR-MS (ESI) values of 10a were same. HR-MS (ESI) m/z: calcd for C26H24NO5 [M]+ 430.1649, found 430.1649.
CURRENT COMMENT: Ok, Thank you.
Point 5: Section 2.3 is not well described to understand the Lineweaver-Burk reciprocal diagram. More detail version of kinetic study of AChE inhibition of 10a should be included.
Response 5: Thanks for the reviewer’s indications. We have rewritten this section.
CURRENT COMMENT: Ok, thank you but can be improved with little more efforts.
Point 6: Table 1: In vitro AChE inhibitory activity of 10b and 13b shows that both are equipotent analogs with different functionalities. A rational for such trend should be provided with special reference to SAR.
Response 6: Thanks for the reviewer’s indications. We have added this section to the SAR
CURRENT COMMENT:
Thank you for authors ‘reply. Please revisit the Table 1. Authors have mentioned that- Overall, sixteen compounds have better inhibitory activities than positive control donepezil (IC50 = 12.06 nM). However, this statement does not correspond to the number of compounds given in Table 1. Apparently only fifteen compounds had better activity than the used standard.
Moreover, author has claimed that-on this basis, when designing new molecules, we introduce acetyl or aminocarbonyl groups that would occupy less space on the pyridine ring for the following reasons. Reviewer feels that authors have designed two series of different analogs which is obvious from the manuscript. However, while performing SAR the authors have mixed the discussion to come up with the finding they had in their minds. Each series have its own characteristics and we cannot compare apples with mangoes simply.
Reviewer would like to recommend revisiting the SAR section and independently describe each series. For an instance, the authors should say that compound 13b (equipotent to 10b) was the most potent in carbamoyl series however 10a was the most potent in acetyl series. After that author should describe why they selected 10a for further investigations and neglected the 13b. Reviewer believes that in addition to point 1 in in round 1, this section has serious flaw.
Furthermore, the authors have described that -By analysis of the substituents on pyridine ring, we found that first series of compounds 10a-10s performed better in activity overall, and for partially identical substitutions, the activity is similar (10b and 13b). But reviewer feels that this statement has no logic and against the facts mentioned in the Table 1. A careful analysis of Table 1 also shows this clearly, for an instance, (10b = 3.32 ± 0.21 vs 13b= 3.16 ± 0.02), (10d = 7.69 ± 0.35 vs 13d= 6.70 ± 0.0), and (10e = 10.85 ± 1.11 vs 13e= 6.28 ± 0.50) etc. This should be corrected or provide logical reasons for the readership of journal.
Point 7: Several linguistic and grammatical mistakes have been detected throughout the manuscript. The reviewer is left with the feeling that manuscript needs language improvement.
Response 7: Thanks for the reviewer’s suggestion. We have reviewed the manuscript carefully and rewritten some sentences now.
CURRENT COMMENT: Ok, thank you but I can see numerous errors in current version.
